

# Downregulation of EB1 impedes Cx43 localization and cardiac conduction after hypothermic ischemia-reperfusion in rats

Chunlei Wen[1,2,*], Rongfeng Yang[1,*], Jing Yi[3], Ying Cao[4], Yuting Song[1], Li An[3], Zijun Wang[1] and Hong Gao[3]

[1] Guizhou Medical University, Guiyang, Guizhou, China
[2] Department of Anesthesiology, Guiyang Maternal and Child Health Care Hospital, Guiyang, Guizhou, China
[3] Department of Anesthesiology, The Affiliated Hospital of Guizhou Medical University, Guiyang, Guizhou, China
[4] Department of Anesthesiology, The Second People's Hospital of Guiyang, Guiyang, Guizhou, China
* These authors contributed equally to this work.

Corresponding author
Hong Gao, anesth@gmc.edu.cn

## ABSTRACT

**Background:** Hypothermic ischemia-reperfusion arrhythmia is a common complication after cardiopulmonary bypass heart surgery, which can lead to hemodynamic disorders and even sudden cardiac death and is still not effectively prevented. This study aims to investigate the role and mechanisms of EB1 in hypothermic ischemia-reperfusion arrhythmia.

**Methods:** 4–6 week old male Sprague-Dawley (SD) rats were randomly assigned to four groups with a control group receiving no treatment. In the treatment groups, the rats received an injection of a negative control adenovirus (AAV9-CON) or an adenoviral vector containing Mapre1 gene (AAV9-EB1) or an equal volume of saline *via* the tail vein. After 4 weeks, untreated rat hearts underwent continuous isolated heart perfusion for 5 min, while the treatment groups were subjected to Langendorff isolated heart ischemia-reperfusion. The multi-electrode array (MEA) technique was used to measure the conduction heterogeneity of rat heart, evaluating the protective effects of EB1 overexpression against reperfusion arrhythmias. Additionally, histological staining and western blotting were used to explore the potential pathways by which EB1 exerts its anti-arrhythmic effects, potentially through promoting the localization of connexin 43 (Cx43) to the intercalated discs (IDs). Furthermore, western blot analysis was conducted to assess microtubule stability and evaluate the possible mechanism by which EB1 facilitates the localization of Cx43 to the IDs.

**Results:** Following ischemia-reperfusion, EB1 expression was downregulated, accompanied by a reduction in Cx43. Overexpression of myocardial EB1 reduced the incidence of reperfusion arrhythmias and shortened their duration, which was associated with improved myocardial conduction. Male SD rats injected with AAV overexpressing EB1 had significantly higher levels of both total myocardial Cx43 and gap junction Cx43 after ischemia-reperfusion compared to the non-overexpression groups. Histological staining revealed lateralization of Cx43 in ischemia-reperfusion myocardium, which was corrected by EB1 overexpression. Additionally, EB1 overexpression increased the distribution of Cx43 at the IDs, overall reducing Cx43 remodeling. Moreover, EB1 overexpression can also alleviate microtubule damage

caused by ischemia-reperfusion, which may be an important mechanism for the transport of Cx43 to the IDs.

**Conclusions:** EB1 downregulation following hypothermic ischemia-reperfusion was accompanied by a reduction in gap junction Cx43. EB1 overexpression improved cardiac conduction and reduced reperfusion arrhythmias by promoting Cx43 localization to IDs, facilitating gap junctions (GJs) formation. These findings contribute to the development of new therapeutic targets for reperfusion arrhythmias.

## INTRODUCTION

Hypothermic ischemia-reperfusion arrhythmia is caused by coronary artery reperfusion after myocardial ischemia during cardiopulmonary bypass heart surgery (*Kadric & Osmanovic, 2017*). Hemodynamic disturbances triggered by reperfusion arrhythmia can lead to sudden cardiac death, affecting surgical success and patient prognosis. Abnormal electrical conduction is an important cause of arrhythmia (*De Coster et al., 2023*; *Wang et al., 2006*). It has been confirmed that the normal conduction of the heart and the abnormalities that lead to arrhythmias are directly related to gap junctions (GJs). GJs form conduits between adjacent cells composed of connexin (Cx) subunits, allowing for direct intercellular communication, with connexin 43 (Cx43) being the most common (*Söhl & Willecke, 2004*; *Rodríguez-Sinovas et al., 2021*). Under normal conditions, GJs are primarily located at the intercalated discs (IDs) between cardiomyocytes, and this longitudinal conduction ensures the synchronization of electrical signals between cardiomyocytes, maintaining the heart's normal rhythm and contraction. Several heart diseases, including acute myocardial infarction and heart failure, are associated with GJs remodeling (*Himelman et al., 2020*; *Lillo et al., 2023*; *Sun et al., 2010*), mainly characterized by reduced expression, abnormal distribution, and phosphorylation changes of Cx43 (*Fontes et al., 2012*; *Sánchez et al., 2011*; *Solan & Lampe, 2018*). Ischemia causes the redistribution of Cx43 from IDs to the sides of cardiomyocytes, reducing end-to-end connections between cells and increasing lateral coupling, leading to anisotropic conduction and arrhythmia. Limited evidence suggests that obstacles in delivering Cx43 hemichannels to IDs are a significant cause of arrhythmia (*Macquart et al., 2019*; *Smyth et al., 2010*).

Cx43 is synthesized in the endoplasmic reticulum, modified in the Golgi apparatus, polymerized into hexameric hemichannels, and finally transported to the cell membrane *via* microtubules (*Basheer & Shaw, 2016*; *Thomas et al., 2005*). The Cx43 half-life in cardiomyocytes is only 1–2 h (*Beardslee et al., 1998*; *Laird, Puranam & Revel, 1991*), indicating a high-intensity dynamic renewal process in GJs. This requires the transport of Cx43 to be efficient and stable. Evidence shows that EB1 directly binds to the ends of microtubules, forming a "comet-like" structure that regulates microtubule growth (*Song et al., 2023*; *Tirnauer et al., 2002*). EB1 also connects microtubule ends to

various cell structures, such as the cell cortex (*Akhmanova & Hoogenraad, 2005*; *Mimori-Kiyosue et al., 2005*). EB1 has multiple protein-binding sites, including the p150-glued protein, a component of the dynein/dynactin complex, which can bind microtubules to adhesive junctions at IDs (*Chausovsky, Bershadsky & Borisy, 2000*; *Ligon et al., 2001*), facilitating the targeted delivery of Cx43 to IDs. Some scholars speculate that the reduction of Cx43 at IDs is partly due to the altered transport direction of microtubules to new regions (*Smyth et al., 2012*). There is also evidence that Cx43 may anchor EB1 and modulate the transport of NaV1.5 to the membrane. Any loss of $Na_V1.5$ forward trafficking will likely also affect conduction (*Agullo-Pascual et al., 2014*; *Marchal et al., 2021*). It can be inferred that EB1 plays a crucial role in the normal beating of the heart.

Our previous studies found that myocardial ischemia-reperfusion increases reperfusion arrhythmia and is accompanied by Cx43 redistribution (*Cao et al., 2022*; *Ma et al., 2023*; *Yi et al., 2022*). To further understand the causes of abnormal Cx43 distribution after myocardial ischemia-reperfusion, this study focuses on EB1, which has been extensively studied for its role in regulating microtubule dynamics. However, research on EB1 in reperfusion arrhythmia is limited. Our results show that EB1 is enriched in normal cardiomyocytes but downregulated after ischemia-reperfusion. We hypothesize that increasing EB1 in the myocardium can rescue the redistribution of Cx43 after ischemia-reperfusion, thereby reducing reperfusion arrhythmias. To explore this hypothesis, we injected adeno-associated virus targeting the heart to overexpress EB1 in rats. Our study found that ischemia-reperfusion increased the occurrence of reperfusion arrhythmia, while overexpression of EB1 through adeno-associated virus successfully improved Cx43 localization and cardiac conduction after ischemia-reperfusion, reducing reperfusion arrhythmia.

# MATERIALS AND METHODS

## Animals

Specific pathogen free (SPF) grade male SD rats were purchased from Changsha (SYXK (Xiang) 2023-0002; Tianqin Biotechnology Co., Ltd., Tianjin, China), including 18 rats aged 8–10 weeks and 40 rats aged 4–6 weeks, and housed them at the Translational Medicine Center of Guizhou Medical University. The rats were housed individually and fed a standard rodent diet, maintained under controlled conditions of temperature (22–24 °C) and humidity (45–50%), with a 12-h light/dark cycle. One week of acclimatization feeding was completed prior to experiments. The rats involved in the experiment were confirmed dead after their hearts were removed under anesthesia and the remaining ones were euthanized before the end of experiment program. The rats were euthanized by inhalation of carbon dioxide gas (flow rate: 50–60% of cage volume/min, no prefilled chambers), ensuring that no other animals were present at the execution site. The bodies were properly handled only after the rats were confirmed dead. All animal experiments were conducted in accordance with the Guide for the Care and Use of Laboratory Animals. The Animal Care and Use Committee of Guizhou Medical University (NO. 2303271) reviewed and approved each experiment.

## Isolated heart ischemia-reperfusion model

Rats aged 8–10 weeks were randomly grouped into the continuous perfusion (CP group) or ischemia-reperfusion (IR group) (six rats in each group). The rats were anesthetized with sodium pentobarbital (60 mg/kg) and anticoagulated with sodium heparin (2000 U/100 kg). When the rats reached deep narcosis, a thoracotomy was conducted, and the hearts were rapidly removed and placed in a 4 °C Krebs-Henseleit (K-H) solution. Then the Langendorff heart perfusion technique was used for retrograde perfusion of the heart *via* the aorta. The non-cyclic Langendorff device provided a constant pressure of 70 mmHg at (37 ± 0.5) °C from an oxygen-containing glass reservoir. Rat heart returned to normal sinus rhythm within 3 min of balanced perfusion and maintained a heart rate ≥180 beats per minute were included in this study. After 30 min equilibration perfusion with 37 °C K-H solution, cardiac arrest was induced by 4 °C Thomas solution infusion and maintained for 60 min. During arrest, myocardial protection was achieved by continuous immersion in 4 °C K-H solution, with half-volume replenishment of 4 °C Thomas solution at 30 min arrest interval. Post-arrest reperfusion was performed with 37 °C K-H solution for 30 min. The CP group hearts were continuously perfused with K-H solution for 120 min. The K-H solution (NaCl 120 mM, KCl 4.5 mM, $CaCl_2$ 1.25 mM, $MgCl_2 \cdot 6H_2O$ 1.2 mM, $KH_2PO_4$ 1.2 mM, $NaHCO_3$ 20 mM, $C_6H_{12}O_6$ 10 mM) was oxygenated with 95% $O_2$/5% $CO_2$ and maintained at pH 7.4.

## Animal experimental design

A total of 4–6 week old male SD rats were randomly divided into treatment groups and a non-treatment group (Group C). In the treatment groups, the rats received an injection of a negative control adenovirus (AAV9-CON group) or an adenoviral vector containing Mapre1 (AAV9-EB1 group) ($1 \times 10^{12}$ v.g/rat; Shanghai Gene Chem Co., Ltd., Shanghai, China) or an equal volume of saline (IR group) *via* the tail vein (eight rats in each group). Four weeks later, the treatment group underwent *ex vivo* myocardial ischemia-reperfusion, while the non-treatment group was subjected to only 5 min of continuous perfusion (to expel residual blood from the myocardium). Throughout the study, no rats exhibited infections, rapid weight loss, or sudden death. However, if such conditions arose, euthanasia would be promptly administered to minimize animal distress.

## *Ex vivo* electrophysiological recordings

Once the isolated heart is placed in the perfusion device, electrodes are fixed to the surface of the left ventricle (determined by the anatomical landmarks of the left anterior descending branch, aorta, and atrium). The 64-channel MAP system (EMS64-USB-1003, Oxford, UK) is used to collect electrical conduction at time points $T_0$, $T_1$, and $T_2$ ($T_0$, balanced perfusion for 30 min; $T_1$, reperfusion for 15 min; $T_2$, reperfusion for 30 min). The heart's electrical conduction data is calculated by EMapRecord 5.0 software (MappingLab Ltd., Oxford, UK), including absolute inhomogeneity ($P_{5-95}$) and the inhomogeneity index ($P_{5-95}/P_{50}$).

## Hematoxylin-Eosin staining

Rats cardiac tissues were fixed with 4% paraformaldehyde (BL539A; Biosharp, Hefei City, China) for 36 h, followed by dehydration in a series of alcohol gradients (70%, 80%, 90%, 95%, 100%) for 5 min each. The conventional method of staining involved hematoxylin-eosin (HE) staining. Finally, the sections were observed under an optical microscope (Olympus, Tokyo, Japan).

## Immunohistochemical staining

Myocardial tissues were immersed in a 4% paraformaldehyde solution for 36 h, embedded in paraffin, and cut into 5 μm thick sections. After de-paraffinization and rehydration, sections underwent heat-induced epitope retrieval using an antigen retrieval solution (Servicebio, Hubei, China). The sections were then incubated overnight with primary antibody at 4 °C (Cx43, 1:1500, Sigma-Aldrich, St. Louis, MO, USA, C6219). Subsequently, the sections were incubated at 37 °C for 30 min with biotinylated goat anti-rabbit IgG antibody, followed by a 30-min reaction with Strept Avidin–Biotin Complex at 37 °C. Color development was performed with DAB substrate with hematoxylin counterstaining.

## Immunofluorescence staining

After the completion of *ex vivo* perfusion, fresh rat myocardium was collected and frozen for sectioning, followed by immunofluorescence staining. Sections were washed three times on a shaker with PBS for 10 min each time. To enhance cell membrane permeability and porosity, the sections were incubated in 0.5% Triton X at room temperature for 30 min. Subsequently, the sections were washed again with PBS three times for 10 min each time. The sections were then incubated in 5% bovine serum albumin (BSA) at room temperature for 1 h. Next, the sections were incubated overnight at 4 °C with primary antibodies (Cx43, 1:500, C6219; N-cadherin, 1:1,000, Cell Signaling Technology, Danvers, MA, USA, #14215; MAPRE1, 1:200, Abmart, Shanghai, China, TD13549). The following day, the sections were rinsed in PBS three times for 10 minutes each time, followed by the addition of a diluted mixture of fluorescent secondary antibodies(FITC-Goat Anti-Rabbit IgG (H+L),1:400, proteintech, Rosemont, IL, USA, SA00003-2; CY3-Goat Anti-Mouse lgG (H+L), 1:400, biopm, PMK-014-095M/S). The sections were then kept in a dark room at room temperature for 1.5 h. Myocardial sections were washed on a shaker with PBS three times for 5 min each time. DAPI (Servicebio, Hubei, China) was applied to the sections and incubated for 5 min, followed by three washes for 5 min each. Finally, the sections were sealed with an anti-fade mounting medium. Imaging was performed using a confocal microscope (SpinSR10; Olympus, Toyo, Japan).

## Extraction of tubulin fractions

To separate the tubulin protein components into free and aggregated forms. We homogenized fresh heart tissue in 1 ml of microtube stabilizing buffer and 0.1% Triton X, where the buffer components included 2 mM ethylenediaminetetraacetic acid, 2 mM

ethylene glycol-bis (2-aminoethylether)-N,N,N′,N′-tetraacetic acid, 0.5 mM MgCl$_2$, 20% glycerol, and 0.1 M piperazine-N,N′-bis (2-ethanesulfonic acid, pH 6.8). The homogenate was centrifuged at 100,000 g for 15 min at 4 °C. The supernatant was collected as the free tubulin protein fraction, and the remaining portion was resuspended in lysis buffer and incubated on ice. After 1 h of incubation, it was centrifuged at 100,000 g for 15 min at 4 °C. The supernatant post-centrifugation was kept as the aggregated tubulin protein component.

### RT-qPCR

Total RNA was extracted from 20 mg fresh heart tissues using TRIzol (Invitrogen, Carlsbad, CA, USA). The quality and quantity of the RNA were assessed through agarose gel electrophoresis (1%, 135 V, 25 min) and NanoDrop One$^c$ (701-058108108; Thermo Scientific, Waltham, MA, USA). A total of 1,000 ng of RNA in a volume of 20 µl volume was reverse transcribed into cDNA according to the manufacturer's instructions (Yeasen, Shanghai, China). The quantitative Real-Time PCR analysis was performed on a CFX96 Real-Time PCR system (Bio-Rad, Hercules, CA, USA) using SYBR Green Master Mix kit (Yeasen, Shanghai, China). The following primers were used: Mapre1 forward: 5′-TGCTTGGGTGAAGAGAGG-3′; Mapre1 reverse: 5′-CAGAGATGTGGGCAGGTC-3′; β-actin forward: 5′-TGTCACCAACTGGGACGATA-3′; β-actin reverse: 5′-GGGGTGTTGAAGGTCTCAAA-3′. The PCR reaction was run at 95 °C for 5 min, followed by 40 cycles of 95 °C for 10 s, 57 °C for 20 s, and 72 °C for 20 s. β-actin was used as an internal reference and the expression was calculated by ΔΔCt method.

### Western blotting

Total protein was extracted using RIPA buffer (Solarbio, Beijing, China) and quantified using the BCA Protein Assay kit (Solarbio, Beijing, China). The membrane and cytoplasmic protein extraction kit (KeyGEN BioTECH, Nanjing, China) was utilized to prepare cell membrane and cytoplasmic protein extracts. After denaturation, the proteins were separated with SDS-page (PG112; Shanghai Yaenzyme Biotechnology Co., Ltd, Shanghai, China) and transferred to PVDF membranes(IPVH00010; Merck Millipore, Billerica, MA, USA). Membrane blockade was conducted with the Protein Free Rapid Blocking Buffer (PS108P; Shanghai Yaenzyme Biotechnology Co., Ltd., Shanghai, China) for 10 min at ambient temperature. The primary antibodies were diluted by Antibody diluent (G2025; Servicebio, Hubei, China) and incubated with the membrane at 4 °C overnight. The primary antibodies included Anti-MAPRE1/EB1 antibody (1:1,000; ab308076; Abcam, Cambridge, UK), Anti-Connexin 43 antibody (1:8,000; C6219; Sigma-Aldrich, St. Louis, MO, USA), N-cadherin Mouse mAb (1:1,000; #14215; Cell Signaling Technology, Danvers, MA, USA), GAPDH antibody (1:10,000; 1E6D9; Proteintech, Rosemont, IL, USA), VDAC1/2 antibody (1:1,000; 10866-1-AP; Proteintech, Rosemont, IL, USA), α-Tubulin (DM1A) Mouse mAb (1:1,000; #3873; Cell Signaling Technology, Danvers, MA, USA) and Anti-Sodium Potassium ATPase antibody (1:20,000; ab76020; Abcam, Cambridge, UK). After washing with TBST, the membrane was incubated with HRP-conjugated Affinipure Goat Anti-Mouse lgG (H+L) (1:1,000, China) or

HRP-conjugated Affinipure Goat Anti-Rabbit lgG (H+L) (1:1,000, China) at room temperature for 1 h. Subsequently, the membrane was visualized using the electrochemiluminescence (ECL) reagent (Bio-Rad Laboratories, Hercules, CA, USA) and bands were detected using the ChemiDoc MP Imaging System (Bio-Rad Laboratories, Hercules, CA, USA). The relative protein levels were represented as the ratio of the grayscale values of the target protein to $\alpha$-Tubulin, ATPase, voltage-dependent anion channels (VDAC) or glyceraldehyde-3-phosphate dehydrogenase (GAPDH).

## Statistical analysis

The GraphPad Prism 9.5.2 was used to analyze the data. Statistical analyses were performed by one-way ANOVA between groups and two-way ANOVA at different time points within groups. One-way ANOVA with Benjamini-Hochberg correction was employed to evaluate the differences among multiple groups for the absolute inhomogeneity and inhomogeneity index at $T_0$, $T_1$, and $T_2$. For other one-way ANOVA, Dunnett's multiple comparison test was used. Additionally, two-way ANOVA followed by Tukey's multiple comparison test was utilized to assess the differences among multiple groups. Comparisons between the two groups were performed by an unpaired t-test. *$P < 0.05$, **$P < 0.01$, ***$P < 0.001$ and ****$P < 0.0001$ were considered as statistically significant with mean ± SD.

# RESULTS

## EB1 downregulation and reduced Cx43 in gap junctions in ischemia-reperfusion model

Figure 1A shows the Langendorff *ex vivo* perfusion experimental procedure. To ensure the successful construction of the hypothermic ischemia-reperfusion model, we used HE staining to detect morphological changes in myocardial tissue. As shown in Fig. 1B, the myocardial tissue exhibited varying degrees of cell swelling and nuclear enlargement after hypothermic ischemia-reperfusion. Previous studies have shown that EB1 reduction in myocardial IDs of heart failure patients is accompanied by a decrease in Cx43 in gap junctions (*Smyth et al., 2010*), suggesting that EB1 may be a biological target for improving Cx43 distribution and predicting arrhythmia. Similarly, this study observed a downregulation of EB1 in myocardial tissue following hypothermic ischaemia-reperfusion (Fig. 1C), accompanied by a reduction in membrane Cx43 (intercellular gap junction Cx43) (Fig. 1D).

## Overexpression of EB1 reduces the incidence and duration of hypo-thermic reperfusion arrhythmia

To further understand the role of EB1 in hypothermic ischemia-reperfusion arrhythmia, we overexpressed EB1 in rat myocardium (Fig. 2A). We detected the expression of the EB1 gene through RT-qPCR, confirming successful virus transfection (Fig. 2B). We used 64-channel MEA mapping technology and ECG to detect changes in cardiac conduction and arrhythmia after hypothermic ischemia-reperfusion. The study found that hypothermic ischemia-reperfusion led to arrhythmia (Figs. 2C–2F). Figure 2C shows typical reperfusion

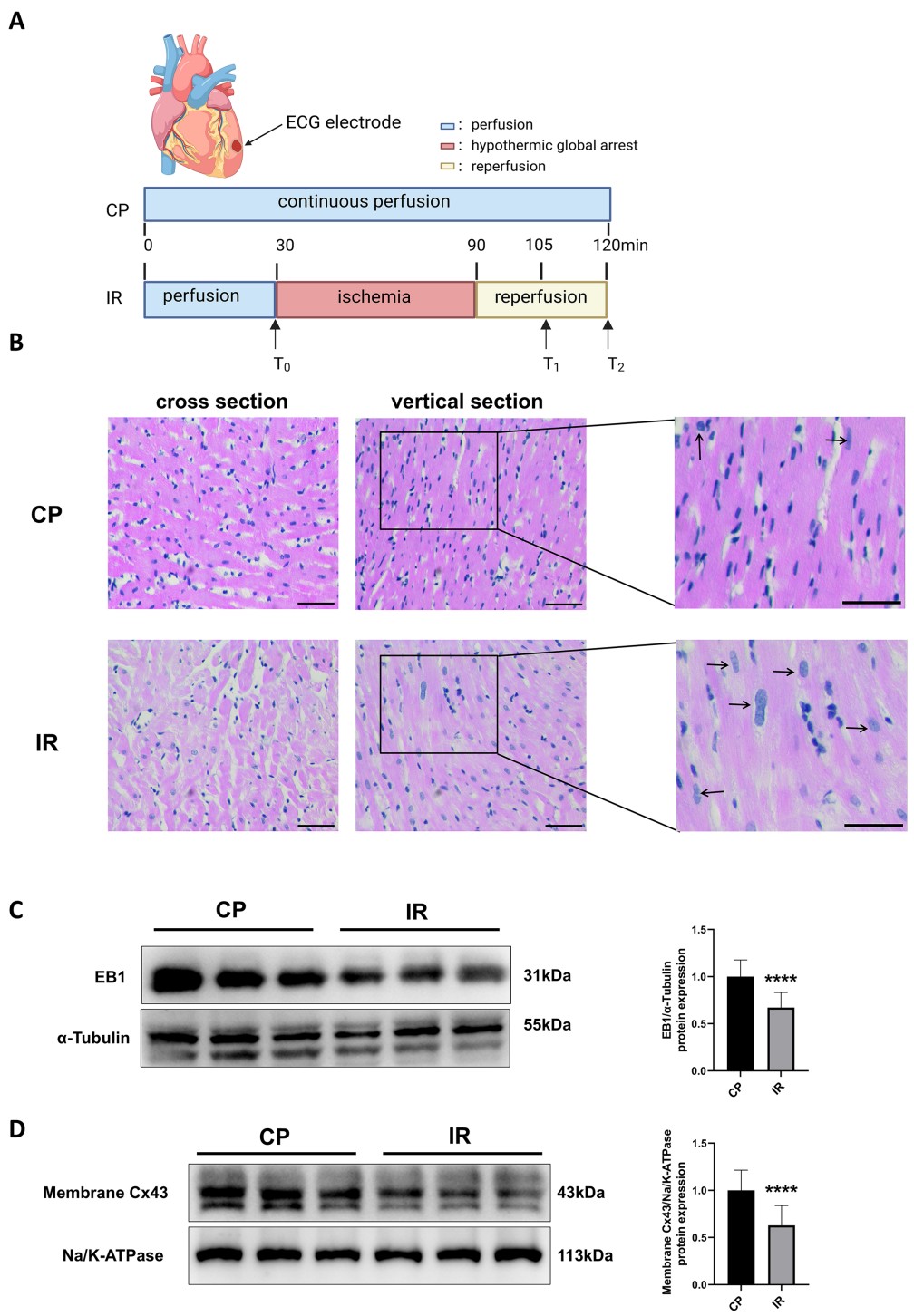

**Figure 1 Downregulation of EB1 under conditions of hypothermic ischemia reperfusion.**
(A) Experiment protocol. CP group was continuously perfused for 120 min and IR group was treated for hypothermia (4 °C) and cardiac arrest for 60 min, followed by 30 min of reperfusion. (B) The histopathological changes were detected by HE staining; scale bar, 50 μm. (C and D) The EB1 and membrane Cx43 protein expression was detected by Western blot assay. α-Tubulin and ATPase were used as loading control respectively. $N = 6$ per group, unpaired t-test. Data are represented by mean ± SD.
****$P < 0.0001$.                               

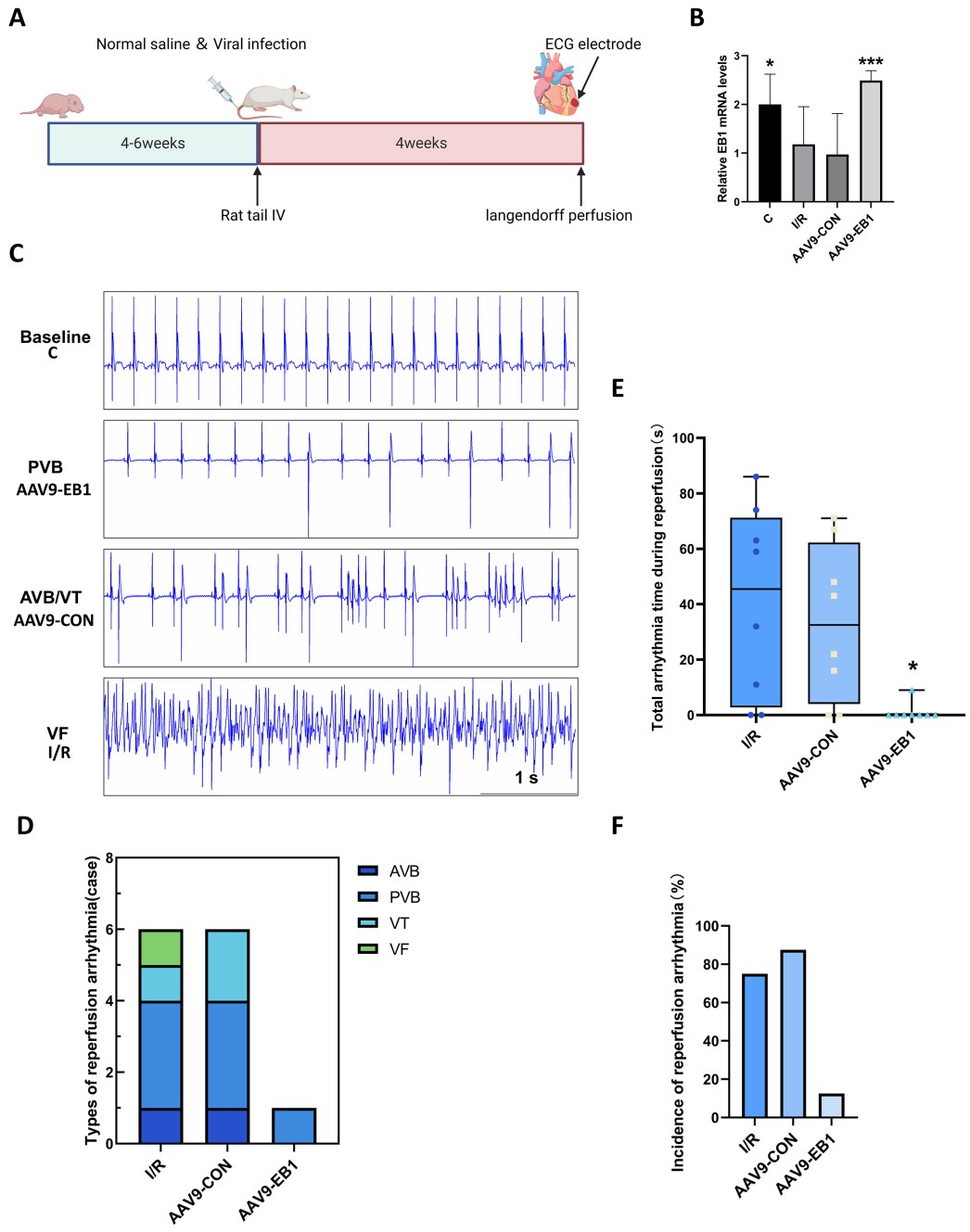

**Figure 2 EB1 reduced hypothermic ischemia-reperfusion arrhythmia.** (A) Experiment protocol for viral transfection. (B) EB1 mRNA levels assessed by RT-qPCR. (C) The typical ischemia-reperfusion arrhythmia. Baseline, The synchronized ECG during balanced perfusion. PVB, ventricular premature beat. AVB, atrioventricular block. VT, ventricular tachycardia. VF, ventricular fibrillation. (D) Types of ischemia-reperfusion arrhythmia. (E) The duration of ischemia-reperfusion arrhythmia. (F) Incidence of ischemia-reperfusion arrhythmia. $N = 8$ per group, one-way ANOVA. Data are represented by mean ± SD. $*P < 0.05$, $***P < 0.001$.

arrhythmia graphs, primarily ventricular premature beats, with atrioventricular conduction block, ventricular tachycardia, and ventricular fibrillation (Fig. 2D). In our study, the frequency and duration of reperfusion arrhythmia were reduced after EB1 overexpression (Figs. 2E and 2F). These results indicate that overexpression of EB1 can reduce reperfusion arrhythmias.

### EB1 improves reperfusion arrhythmia by correcting myocardial conduction after hypothermic ischemia-reperfusion

To further explore the protective mechanism of EB1 in reperfused myocardium, we monitored the electrophysiological changes in hypothermic ischemia-reperfused myocardium in each group. This study used a multi-electrode array mapping technique to further investigate the following abnormalities of myocardial electrical conduction: (1) The increased temporal and spatial dispersion of conduction; (2) inhomogeneity indices.

The study found that hypothermic ischemia-reperfusion caused changes in the location and time of electrical activation. In the IR and AAV9-CON groups, compared with $T_0$, we observed changes in activation locations and prolonged activation times at $T_1$ and $T_2$ after reperfusion from the conduction activation maps, whereas the AAV9-EB1 group remained stable (Fig. 3A). These findings suggest that post-reperfusion electrophysiological remodeling, characterized by altered conduction velocity and/or pathological propagation pathways, may underlie the development of spatiotemporal activation heterogeneity across myocardial regions. Statistical analysis of heart rate parameters revealed no significant differences across the observed time points ($P > 0.05$) (Fig. 3B). We postulate that there are two possible reasons for the absence of statistically significant differences in heart rate changes. First, arrhythmias do not necessarily occur precisely at the acquisition time points of $T_0$, $T_1$, and $T_2$. Second, as we observed, arrhythmias mainly occur within 15 min after ischemia-reperfusion, and they become less common at the $T_1$ time point.

Myocardial electrical conduction abnormalities serve as pathophysiological biomarkers for reperfusion arrhythmogenesis, where increased conduction heterogeneity establishes the electrophysiological substrate for re-entrant circuits. Notably, conduction heterogeneity metrics, specifically absolute conduction heterogeneity and heterogeneity index, provide quantitative assessment of regional tissue susceptibility to arrhythmic re-entry (*Lammers et al., 1990*). Figure 3A is the activation map, while Fig. 4A is the dispersion map at that time point, with higher dispersion indicating greater cardiac electrical conduction inhomogeneity. In the IR and AAV9-CON groups, compared with $T_0$, we observed increased dispersion at $T_1$ and $T_2$ after reperfusion from the dispersion maps, while the AAV9-EB1 group remained stable (Fig. 4A), which was consistent with the changes in conduction activation time in Fig. 3A. Additionally, we recorded and statistically analyzed the absolute inhomogeneity ($P_{5-95}$) and inhomogeneity index ($P_{5-95}/P_{50}$) of conduction. In the IR and AAV9-CON groups, the inhomogeneity of rat myocardium increased after hypothermic ischemia-reperfusion (Figs. 4H and 4I). However, compared with the IR group, the inhomogeneity index and absolute inhomogeneity of myocardial conduction after reperfusion ($T_1$ and $T_2$) were significantly

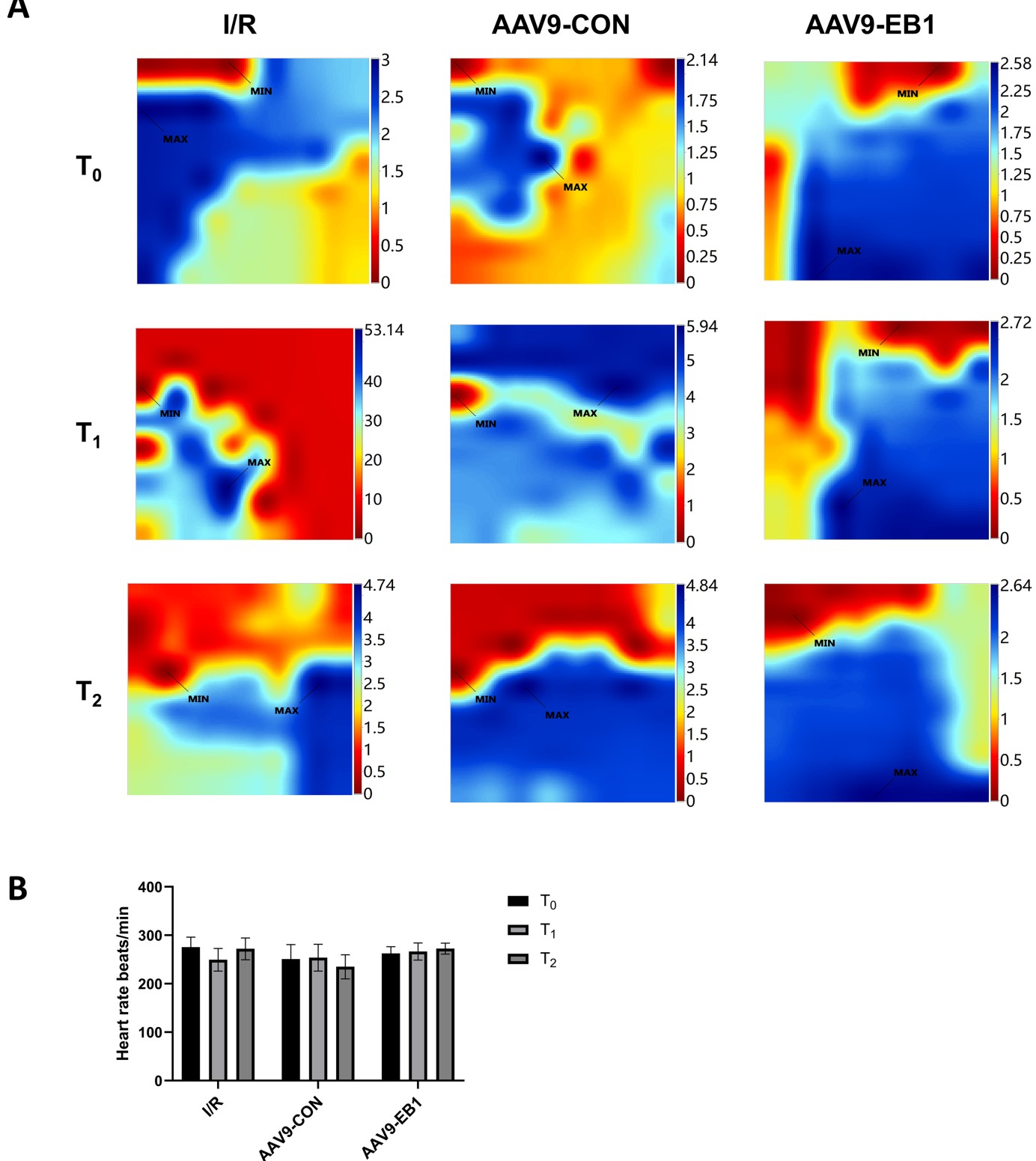

**Figure 3 EB1 improved myocardial electrical activation during ischemia-reperfusion.** (A) Classic activation maps were obtained in the three groups at $T_0$, $T_1$ and $T_2$. (B) Comparison of heart rate at different time points for each group. MAX and MIN are the maximum and minimum values of local activation time respectively. Data expressed as mean ± SD ($n$ = 8/group). Two-way ANOVA revealed no statistically significant intergroup differences ($P$ > 0.05 for all comparisons).

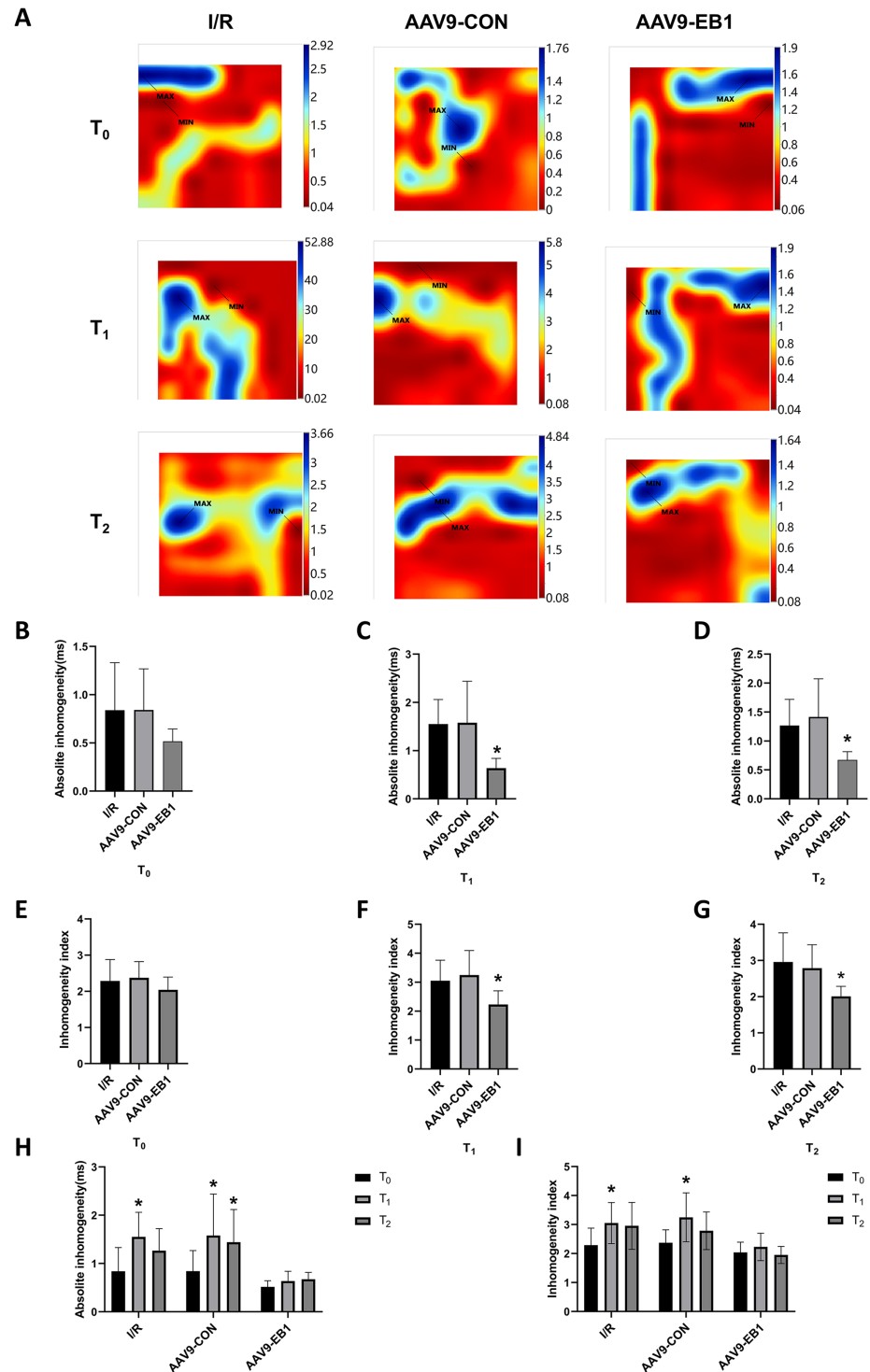

**Figure 4 EB1 improved cardiac conduction inhomogeneity during ischemia-reperfusion.** (A) Representative inhomogeneity maps were acquired in the three groups at $T_0$, $T_1$ and $T_2$. (B–D) The absolute inhomogeneity respectively at $T_0$ (B), $T_1$ (C), and $T_2$ (D), *vs group IR. (E–G) The inhomogeneity index respectively at $T_0$ (E), $T_1$ (F), and $T_2$ (G), *vs group IR. (H and I) Comparison of absolute inhomogeneity and inhomogeneity index at different time points for each group, *vs group IR. MAX and MIN are the maximum and minimum values of inhomogeneity respectively. $N = 8$ per group, one-way ANOVA or two-way ANOVA. All data are represented by mean ± SD. *$P < 0.05$.

reduced in the AAV9-EB1 group (Figs. 4C, 4D, 4F, 4G). These results suggest that overexpression of EB1 improved cardiac conduction.

## Overexpression of EB1 rescues Cx43 after hypothermic ischemia-reperfusion

Our previous research found that after ischemia-reperfusion, both EB1 and gap junction protein Cx43 were reduced. Next, we investigated the reasons why EB1 improved cardiac conduction after ischemia-reperfusion. Therefore, we used different experimental methods to investigate the relationship between EB1 and Cx43 in myocardial ischemia-reperfusion. In the fluorescence images (Fig. 5A), we observed that EB1 was distributed vertically towards N-cadherin and tended to localize at N-cadherin, which was consistent with the distribution of EB1 along microtubules. We hypothesize that EB1 concentrates towards N-cadherin to deliver Cx43 to the IDs, promoting the distribution of Cx43 at the IDs. EB1 was significantly expressed in the AAV9-EB1 group (Figs. 5A and 5B), while N-cadherin was not affected by EB1 overexpression. We also observed that the reduced Cx43 in cardiomyocytes after ischemia-reperfusion was prevented by the overexpression of EB1, rescuing the loss of Cx43 after ischemia-reperfusion (Fig. 5B).

Our previous sequencing analysis results indicated the involvement of microtubules in ischemia-reperfusion (Huang et al., 2024). Microtubules can assist in the transport of Cx43, as well as a multitude of other proteins including $Na_V1.5$. Therefore, we also assessed the relationship between EB1 and microtubule stability. Western blot analysis showed that, compared to the control group, the polymerization of microtubules increased and depolymerization decreased in the EB1 overexpression group, indicating that EB1 overexpression can enhance the stability of microtubules after ischemia-reperfusion (Figs. 5C and 5D).

## EB1 promotes Cx43 localization to IDs and reduces Cx43 lateralization

To further elucidate whether EB1 overexpression improves reperfusion arrhythmia by rescuing the loss of gap junction Cx43 at IDs, we used IHC and IF to detect the distribution of Cx43 to evaluate the role of EB1. In both IHC and IF, we observed that, compared to the IR and AAV9-CON groups, Cx43 is more concentrated at the GJs in the C group and AAV9-EB1 group, with intact morphology and orderly arrangement (Figs. 6A and 6B). In the immunohistochemical images, we noted the lateralization of Cx43 (Fig. 6A, indicated by arrows). Given that N-cadherin is present at GJs and exhibits high stability (Kostetskii et al., 2005), we used N-cadherin as a control in the IF detection experiments. In the fluorescence images, we also observed prominent lateralization of Cx43 in the IR and AAV9-CON groups. In the right-sided insets, the upper image shows the co-localization of Cx43 with N-cadherin, representing the GJs, while the lower image, indicated by arrows, shows the lateralization of Cx43 (Fig. 6B), consistent with the immunohistochemical (IHC) findings. In addition, we measured myocardial membrane Cx43 levels by immunoblotting. Compared with the IR group, membrane Cx43(gap junction Cx43 in cardiomyocytes) was significantly increased in rat cardiomyocytes overexpressing EB1 (Fig. 6C). These results indicate that EB1 promotes

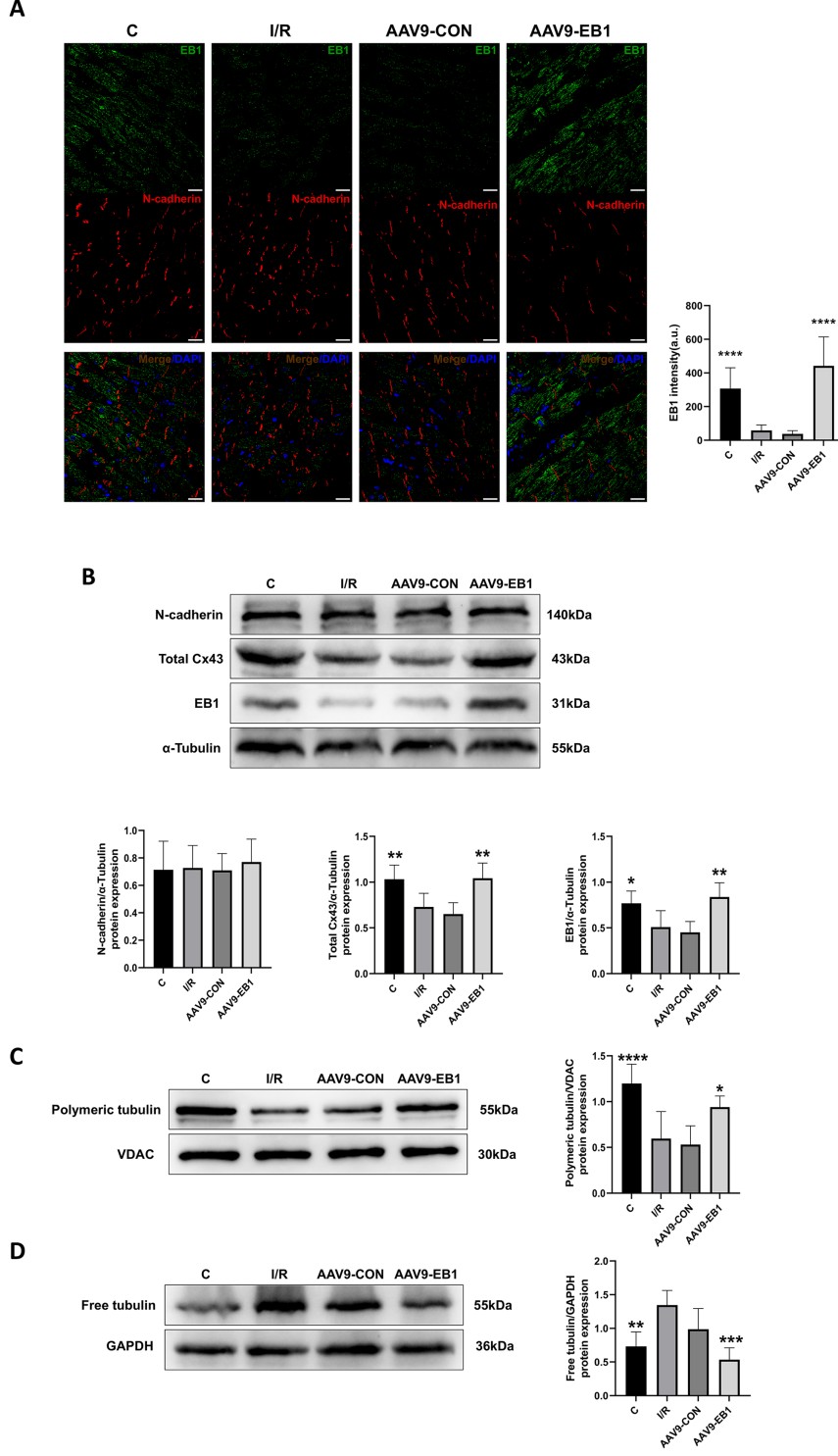

**Figure 5 Overexpression of EB1 suppressed the reduction of Cx43 during ischemia-reperfusion.**
(A) The location and expression of EB1 by immunofluorescence assay. N-cadherin was used as a marker for the IDs. N-cadherin (red), EB1 (green), nuclei (DAPI, blue). Scale bar, 20 μm. (B–D) Immunoblotting and quantification for N-cadherin, Cx43, EB1, polymeric tubulin and free tubulin. α-Tubulin, VDAC and GAPDH were used as loading control respectively. $N = 6$ per group, one-way ANOVA. All data are represented by mean ± SD. *$P < 0.05$, **$P < 0.01$, ***$P < 0.001$, ****$P < 0.001$.

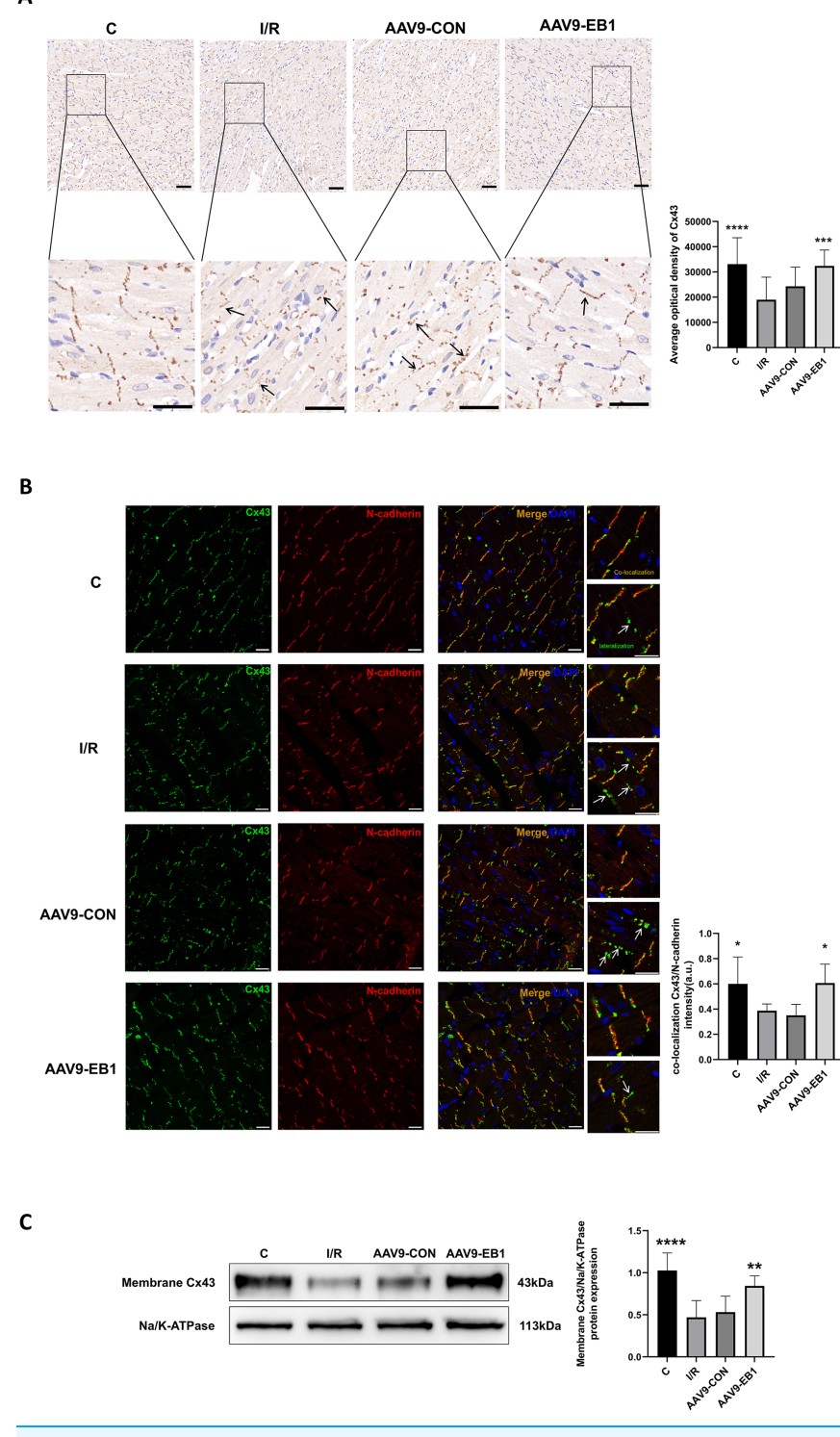

**Figure 6 EB1 can promote the distribution of Cx43 to IDs and reduce the lateralization of Cx43.**
(A) The distribution and expression of the Cx43 protein was recorded by IHC. $N = 5$ per group. Scale bar, 50 μm. Brown shows the expression of the Cx43 protein. Blue shows the nuclei of cardiomyocytes.
(B) Cx43 co-localization with N-cadherin were assessed by immunofluorescence assay and quantified with ImageJ. N-cadherin (red), Cx43 (green), nuclei (DAPI, blue). $N = 5$ per group. Scale bar, 20 μm.

**Figure 6 (continued)**
(C) Immunoblotting and quantification for membrane Cx43. Na/K-ATPase were used as loading control. $N = 6$ per group, one-way ANOVA. All data are represented by mean ± SD. *$P < 0.05$, **$P < 0.01$, ***$P < 0.001$, ****$P < 0.001$.               

the localization of Cx43 at gap junctions following hypothermic ischemia-reperfusion and reduces the lateralization of Cx43.

## DISCUSSION

The results collected from this study indicate that during myocardial ischemia-reperfusion, the EB1 protein plays an important role in reducing the remodeling of Cx43 by regulating its localization, which in turn affects reperfusion arrhythmias. Studies consistently show that the occurrence of ischemia-induced electrical uncoupling and arrhythmias is closely related to Cx43 channel remodeling, including increased internalization, lateralization, and degradation of Cx43 (*Martins-Marques et al., 2015*). Therefore, identifying the factors affecting the abnormal distribution of Cx43 during hypothermic ischemia-reperfusion may open new avenues for the development of arrhythmia prevention strategies.

Disturbances in the generation or propagation of cardiac action potentials can trigger arrhythmias, manifested as abnormalities in heart rate or rhythm (*Nerbonne & Kass, 2005*). GJs, primarily composed of Cx43, play a vital role in normal heartbeats by facilitating the rapid transmission of action potentials between adjacent cells. Cx43 protein has a relatively short half-life, and the rapid synthesis and timely delivery of these proteins to IDs are crucial for maintaining coupling and electrical conduction in cardiomyocytes. Impaired transport of Cx43 can lead to serious complications in diseased hearts, such as arrhythmias associated with sudden cardiac death (*Kalcheva et al., 2007*; *Smyth et al., 2010*). EB1 has been found to be involved in the targeted delivery of Cx43 to adherens junctions by tethering microtubule plus ends at adherens junctions in IDs, facilitating the delivery of Cx43-containing vesicular cargo (*Smyth et al., 2010*). We are interested in whether EB1 plays this role during hypothermic ischemia-reperfusion and whether it can reduce the incidence of reperfusion arrhythmias.

Myocardial ischemia increases the internalization of Cx43 hemichannels at GJs, which are further degraded by the ubiquitin-proteasome system (*Martins-Marques et al., 2020*; *Smyth et al., 2014*), leading to a decrease in Cx43. Our data also confirm the reduction of Cx43 in gap junctions after hypothermic ischemia-reperfusion, and we also observed the loss of EB1. Studies have shown that EB1 is involved in the localization of Cx43 to IDs, contributing to the transport of Cx43 and possibly other functions. We hypothesize that if the loss of EB1 prevents its involvement in Cx43 transport, the internalized Cx43 at IDs will not be replenished, leading to a reduction in GJs and impaired electrical conduction in the heart. Therefore, we used adenovirus-mediated overexpression of EB1 or control adenovirus to target and transfect adult rat myocardial tissues. The results showed that EB1 reduced reperfusion arrhythmias by correcting myocardial electrical conduction after hypothermic ischemia-reperfusion, which was related to EB1 rescuing Cx43. Microtubule damage can be induced in the early stages of ischemia (*Vandroux et al., 2004*), and since

EB1 can regulate microtubule dynamics, we also explored EB1's protective effect on microtubules after ischemia-reperfusion. The results showed that EB1 increased the stability of microtubules in reperfused cardiomyocytes. Intact microtubules are crucial for the transport of Cx43. It is not difficult to infer that under ischemia-reperfusion conditions, EB1 downregulation leads to microtubule depolymerization, while overexpression of EB1 reduces microtubule catastrophe events and ensures microtubule integrity. Additionally, in confocal fluorescence images, overexpressed EB1 tended to the IDs, suggesting that in addition to protecting microtubules, EB1 may also assist in further delivering Cx43 *via* microtubules.

In addition to the observed decrease in EB1 levels after ischemia-reperfusion, we also noticed lateralization of Cx43. In ischemic heart disease, Cx43 is lateralized to the non-intercalated disc sarcolemma and scattered throughout cardiomyocytes bordering infarct zones (*Smith et al., 1991*). We also observed Cx43 lateralization in myocardial tissues overexpressing EB1, but it was more pronounced in the non-EB1 overexpression group after ischemia-reperfusion. Mechanistically, both structural remodeling of gap junctions at IDs and pathological lateralization of Cx43 create a pro-arrhythmic milieu through impaired electrical coupling. This electrophysiological derangement was experimentally demonstrated in our animal models, with control groups exhibiting significant arrhythmic events including AVB, PVB, VT and VF using standardized arrhythmia classification criteria. While contemporary research increasingly focuses on lateralized Cx43-mediated arrhythmogenesis (*Kieken et al., 2009*; *Martins-Marques et al., 2020*), our findings highlight that ischemic-phase Cx43 redistribution constitutes the critical priming factor determining reperfusion-related electrical instability.

Furthermore, we learned that EB1 is a member of the microtubule plus-end tracking proteins. It not only promotes microtubule growth but also delivers Cx43 vesicles toward the IDs (*Nehlig et al., 2017*; *Shaw et al., 2007*). Research has shown that when EB1 dissociates from the microtubule ends, the delivery of Cx43 to GJs decreases, confirming that the displacement of EB1 limits the formation of GJs (*Smyth et al., 2010*). Additionally, in myocardial cells of rats with right ventricular pressure overload, significant lateralization of Cx43 was observed, along with the loss of EB1 enrichment at the IDs (*Chkourko et al., 2012*). Our previous studies have found that ischemia-reperfusion increases the lateralization of Cx43, which is associated with the occurrence of reperfusion arrhythmias (*Ma et al., 2023*; *Yi et al., 2022*). This study further explored the possible mechanisms by which EB1 reduces reperfusion arrhythmias and found that EB1 promotes the localization of Cx43 to IDs and reduces the lateralization of Cx43. We propose that the mechanism of abnormal Cx43 distribution after hypothermic ischemia-reperfusion may be related to the reduced delivery of Cx43 by EB1. On one hand, the downregulation of EB1 further aggravates microtubule damage, and the damaged microtubules cannot dock with the plasma membrane, preventing Cx43 from being delivered to the membrane to form GJs. On the other hand, the reduced total amount of EB1 results in fewer EB1-microtubule end connections, which in turn decreases Cx43 delivery, ultimately preventing Cx43 from being localized at the IDs.

Although this study focuses on the role of EB1 in hypothermic ischemia-reperfusion arrhythmia through regulating Cx43 localization and microtubule stability, further exploration of other potential mechanisms is warranted. Microtubule, as the central cytoskeletal framework for intracellular transport, are not only involved in the targeted delivery of Cx43 but may also influence the trafficking of other membrane proteins, such as ion channels (*e.g.*, Nav1.5, Kir2.1), adhesion molecules (*e.g.*, N-cadherin, β-catenin), or additional connexins (*e.g.*, Cx40, Cx45), all of which play critical roles in electrical conduction and intercellular coupling. Prior studies have demonstrated that aberrant microtubule dynamics disrupt Nav1.5 channel distribution, thereby impairing action potential upstroke velocity and conduction efficiency (*Marchal et al., 2021*). Additionally, interactions between adhesion molecules and microtubules may indirectly modulate gap junction functionality by regulating the stability of intercalated disc structures (*Shaw et al., 2008*). While this study primarily emphasizes Cx43's contribution to ventricular electrical conduction, the pathogenesis of atrioventricular block may involve dysregulation of other connexins (*e.g.*, Cx40, Cx45) or ion channels (*Hagendorff et al., 2001*; *Cruz et al., 2024*; *Li et al., 2021*). Although EB1 overexpression reduced ventricular arrhythmias (*e.g.*, ventricular fibrillation) in our experiments, the observed improvement in atrioventricular block could be attributed to enhanced trafficking efficiency of Cx40/Cx45 due to stabilized microtubules or to the regulation of other ion channels. This highlights the limitations of focusing solely on Cx43 in the current investigation. Consequently, EB1 downregulation-induced microtubule impairment may exacerbate electrical heterogeneity through multiple pathways, and the cardioprotective effects of EB1 overexpression may partially arise from its coordinated regulation of other critical membrane proteins. Future studies should systematically analyze EB1's impact on the localization of additional membrane proteins to comprehensively elucidate its pleiotropic mechanisms in ischemia-reperfusion arrhythmia.

## CONCLUSIONS

In summary, the present study demonstrated a reduction in EB1 expression during ischemia-reperfusion. EB1 overexpression rescued the Cx43 mislocalization at the IDs during ischemia-reperfusion, thereby reducing arrhythmia. EB1 also enhanced microtubule stability during ischemia-reperfusion, further promoting the Cx43 localization to the IDs, facilitating electrical conduction, and ultimately reducing arrhythmia. However, as microtubules serve as a shared pathway for trafficking diverse membrane proteins, their dysfunction may disrupt electrophysiological homeostasis *via* multiple routes. EB1 may synergistically regulate the distribution of ion channels or adhesion molecules, indirectly reducing conduction heterogeneity. Moreover, the prevention of atrioventricular block might involve restored trafficking of other connexins rather than solely relying on Cx43 repositioning. Thus, future research should extend to additional membrane protein targets and explore EB1's region-specific roles. While our findings provide novel insights into ischemia-reperfusion arrhythmia mechanisms, the complexity of the EB1 regulatory network suggests that multi-target combination intervention strategies may hold greater clinical potential.

### Funding

This work was supported by the Guizhou Provincial Science and Technology Projects (Grant No. Qiankehejichu ZK[2024] yiban596). The funders had no role in study design, data collection and analysis, decision to publish, or preparation of the manuscript.

### Grant Disclosures

The following grant information was disclosed by the authors:
Guizhou Provincial Science and Technology Projects: Qiankehejichu ZK[2024] yiban596.

### Competing Interests

The authors declare that they have no competing interests.

### Author Contributions

- Chunlei Wen conceived and designed the experiments, performed the experiments, analyzed the data, prepared figures and/or tables, authored or reviewed drafts of the article, and approved the final draft.
- Rongfeng Yang performed the experiments, prepared figures and/or tables, and approved the final draft.
- Jing Yi performed the experiments, prepared figures and/or tables, and approved the final draft.
- Ying Cao conceived and designed the experiments, analyzed the data, authored or reviewed drafts of the article, and approved the final draft.
- Yuting Song performed the experiments, prepared figures and/or tables, and approved the final draft.
- Li An conceived and designed the experiments, authored or reviewed drafts of the article, and approved the final draft.
- Zijun Wang conceived and designed the experiments, authored or reviewed drafts of the article, and approved the final draft.
- Hong Gao conceived and designed the experiments, analyzed the data, authored or reviewed drafts of the article, and approved the final draft.

### Animal Ethics

The following information was supplied relating to ethical approvals (*i.e.*, approving body and any reference numbers):

The Animal Care and Use Committee of Guizhou Medical University provided full approval for this research (NO. 2303271)

### Data Availability

The raw measurements are available in the Supplemental Files.
## Supplemental Information

Supplemental information for this article can be found online at http://dx.doi.org/10.7717/peerj.19276#supplemental-information.

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
