# Peer review of "Downregulation of EB1 impedes Cx43 localization and cardiac conduction after hypothermic ischemia-reperfusion in rats"

_PeerJ, doi:10.7717/peerj.19276_

## Round 0.1 · original submission · Major Revisions

Please address concerns of all reviewers and revise the manuscript accordingly.

Reviewer 1 ·

Basic reporting

The manuscript includes a comprehensive review of relevant literature, offering sufficient background and context for readers. The structure of the article follows a professional format, and the use of figures and tables effectively supports the presentation of data. Additionally, the raw data has been made available, which enhances the transparency and reproducibility of the research. The manuscript is self-contained, with results that are directly relevant to the hypotheses posed at the outset of the study.

However, while the content is strong, the manuscript requires thorough proofreading. There are several noticeable grammatical errors scattered throughout the text that detract from the clarity and professionalism of the writing. These issues should be addressed to ensure that the manuscript meets the highest standards of academic writing. Despite these minor issues, the overall quality of the research is acceptable, and with some refinement in the language, the manuscript has the potential to make a significant impact in the field.

Experimental design

No comment

Validity of the findings

No comment

Additional comments

ABSTRACT: Background: state a brief background on why EB1is chosen to be studied in the abstract
Methods: The experimental protocols need to be re-written. There are 2 age group. I suggest summarize the main experiment only and mention the IR model construction briefly. How many groups of animal? What was administered to the animal in each group. Which group act as control? Is it wild-type or transgenic animal? Adeno-associated virus (AAV) is a general vector. What was the purpose? What gene it carries? Is it for knockdown of overexpression

INTRODUCTION: Spell out GJs at IDs at the first mention in Introduction.

MATERIALS AND METHODS: Line 110: avoid using ‘we’ in the sentence (specific pathogen free (SPF) grade male SD rats were purchased from….)
Line 125-151: IR model construction and the animal experiment involving injection of AdV are better separated into two subtopics.
Line 209-216: rephrase the sentences to reporting format

RESULTS: Line 287: state the labelling for these groups in the figure 2 legend; adeno-associated virus overexpressing EB1(AAV9-EB1), empty virus (?), and negative control agents (?). some western blot figures do not have labelling

Reviewer 2 ·

Basic reporting

A. Overall, clear and professional English has been used throughout the manuscript. However, there are some paragraphs that could benefit from sentence restructuring/other modifications to improve clarity. For example:
Line 31-34: Abstract methods: The authors need to explicitly state that the study uses an ex vivo model. Current text suggests in vivo model.
Line 51: Restructure sentence to improve clarity for “ischemia-reperfusion-induced microtubule damage”.
Line 84: Restructure sentence, too many “and”.
Line 100-101: Restructure sentence.
Line 148-149: Remove “adverse events” to improve clarity.
Line 157: Remove “and”, start a new sentence.
Line 148-151: Irrelevant for current experiment, suggest remove description entirely.
Line 209-210: Please correct grammar and sentence.
Line 212-216: Please correct grammar and sentences.
Line 227-230: Sentence incomplete.
Line 232-233: Revise sentence and grammar for clarity.
Line 244-245: Revise sentence, correct misspelled word.
Line 251-253: Grammar incorrect, remove last sentence for clarity.
Line 263-265: Consider moving “compared to….” to the end of sentence. Remove “indicated by..” in text because this should be self-explanatory in the figure legend.
Line 266: Consider removing sentence to improve clarity.
Line 276-277: Remove “an” and “global” for clarity.
Line 286-288: Sentence redundant with methods section. Suggest to just state EB1 overexpression.
Line 292: remove “occurrence” for clarity.
Line 292-294: Suggest remove “even…. also occurs” to improve clarity.
Line 310: Suggest remove for clarity. All arrhythmias have an abnormal electrical activity, not just reperfusion arrhythmia.
Line 314-316: Revise sentence for clarity. Use Arabic numbers e.g. (1) and (2) to describe conduction abnormalities.
Line 365-366: Revise sentence for clarity. What are the authors looking for?
Line 507-509: Remove last sentence. Revise remaining sentence to state that ‘there still requires further investigation on….’.
Line 436-462: Please restructure sentences in these paragraphs to improve clarity and coherence. Please remove phrases such as “it is not difficult to infer that…” or long sentences such as “… and since EB1 can…”.
Line 473-475: Revise sentence to improve clarity. Remove “…that needs to be noted”.
Line 507-509: Remove last sentence “This will depend on…”. I suggest stating that the mechanisms still require further investigation.


B. Overall, sufficient background/context has been provided throughout the manuscript. However, some terminologies should either be removed or replaced with more appropriate ones to improve clarity. Others are misspelled, unnecessarily abbreviated or have incorrect formatting. For example:
Line 70: Remove “cardiomyopathies”. Modify sentence to reflect accurate knowledge. Cardiomyopathy is heart muscle disease. Acute myocardial infarction and heart failure are not forms of cardiomyopathy. Acute myocardial infarction can be the etiology, whereas heart failure can be the consequence of cardiomyopathy.
Line 82: Previous sentences do not provide adequate information to support this statement. I suggest add brief information regarding Cx43 transport before this sentence. Alternatively, the authors could modify the current sentence to sound less conclusive.
Line 84: Please restructure sentence to improve clarity.
Line 102-103: Sentence will be much clearer if “confirming our assumption” is removed.
Line 125: Consider replacing the term “construction/constructing” with “model/modelling” or “simulation/simulating” throughout the manuscript.
Line 161: Provide long terminology for “HE”.
Line 175-195: Please correct all misspelled words such as “comple”, “mosue”, etc throughout the manuscript. Please also ensure all latin words such as “in vivo, ex vivo” are written in italics.
Line 221: Consider adding the following “…calculated using the delta delta Ct method”.
Line 232-233: Remove unnecessary abbreviations in text, such as “min” (should be minutes).
Line 294: Consider replacing “incidence” with “frequency” to describe the number of arrhythmia occurrence.
Line 295: Suggest change “improved” with ‘measurable’ verbs e.g. decreased, reduced.
Line 465: remove “immunodetectable”.
Line 504: Provide long terminology for “RA”.


C. The authors have shown effort to cite relatively recent literature references (within 10 years prior), although older references published in the 1990’s and 2000’s should still be replaced with more recent publications.

D. Article structure, figures and tables are well presented. Some figures and figure legends still require additional descriptions to improve clarity. For example:
Line 278-281:
- Fig.1 legend: Instead of “HE staining of…” and “Immunoblotting and quantification...”, I suggest provide brief description on what is shown by the images/graphs. For example: “Protein expression of….”. Please also check the punctuations and uppercase letters.
- Fig.1 legend: Please also describe the arrows in HE images.
- Fig.1 legend: Why are there 3 protein bands in blotting images? If they are biological repeats, please indicate in figure legend.
Line 303-305: Fig.2 legend: Please be consistent regarding the use of uppercase letter in p values.

Experimental design

A. Research gap and research question are well defined and relevant. Experimental methods have been described comprehensively. However, the authors did use rats from different ages for ischemia-reperfusion (I/R) only (8-10 weeks; adult rats) and I/R+AAV-EB1 overexpression (4-6 weeks; teen-young adult). These rats may have differences in cardiac physiology/histology. It would be more informative if the authors provide an explanation regarding this or include data that show no differences between the two age groups.

B. It is advised that authors provide only succinct details of lab products and remove unnecessary uppercase letter in product names. For example:
Change “Shanghai Gene Chem Co. Ltd, Shanghai, China” to “Shanghai Gene Chem, China’.
Change “BL539A, Biosharp, Beijing, China” to “Biosharp, China”, etc.
Change “NANODROP ONE c” to “NanoDrop One”, etc.

Validity of the findings

A. Overall, the authors have provided adequate data to support their hypotheses. There are instances where they have written conclusions that are “too bold”. In this case, more data or evidence from the literature should be incorporated in that paragraph. Alternatively, I advise that conclusions be limited to supporting results described in the text. For example:
Line 296-297: The phrase “crucial role” is not supported by sufficient elaboration of evidence in prior sentences. I suggest incorporating data showing that knockdown of EB1 is accompanied by increased frequency/duration of ischemia-reperfusion arrhythmia.
Line 338-340: This sentence implies a direct causal relationship between EB1 and ischemia-reperfusion arrhythmia. Data in text do not provide sufficient support for this claim. I suggest to modify conclusion to suit current data.

B. The authors have shown effort to ensure that data are robust and statistically sound. However, additional description or figures could be provided to enhance the manuscript. For example:
Line 317-320: Please elaborate on the definition of “location” and “time” in the text. Describe the different time points T0-T2 in text, figure legends and methods section.

Line 322:
- Fig. 3B-D and 3E are redundant. Is it correct that B-D are split versions of E? Please also add title above graphs in B-D to describe the time points T0-T2.
- If T0-T2 represent the time before ischemia-reperfusion, after ischemia and after reperfusion, respectively: does it mean that AAV9-EB1 group is minimally affected or unaffected by ischemia-reperfusion? Are there any data to support this?

Fig. 3F: Graph suggests that heartbeats of all groups are normal. This is in contrast with ECG findings in Fig. 2C which shows AV block (slower heart rate) and Ventricular Fibrillation (faster heart rate). Can you explain this discrepancy?
Line 327-340: Based on data in Fig. 3A and 4A, it will be more helpful to readers if more explanation is provided in text, e.g. focal regions of activation, areas of conduction delay, the extent of dispersion, etc. Please also correct misspelled word “Absolite” in Fig. 4.
Line 384:
- Fig. 5A: What are the blue artefacts in “Merged” images? Are they nuclei stained with DAPI? If so, please add DAPI layer in image and specify this in the figure legend.
- Fig. 5B-D: If protein bands represent each group in the bar graph, please add the group’s name above each protein band.

Line 397:
- Fig. 6A: In IHC images, it is unclear whether Cx43 is more abundant in a particular group, since there is no quantification. I suggest remove the following descriptions “abundant distribution” and “intact morphology” from the text. Alternatively, the authors could describe the images by comparing spatial distribution and position of Cx43 (e.g. lateralization) in different groups—which can also be confirmed by the immunofluorescence images.
- Fig. 6C: Please replace the term “membranal Cx43” with “membrane Cx43” or “surface Cx43” in all figures throughout the manuscript.

Additional comments

Overall, this manuscript is thoroughly written. It provides new insights in the role of EB1 in arrhythmia due to hypothermic ischemia-reperfusion injury. Revisions regarding the use of English language and correct formatting will further enhance the readability of this manuscript.

Reviewer 3 ·

Basic reporting

Some references are oddly selected, and more original/key papers should be cited. When using review this should be made clear. Some examples are given in specific comments.

Experimental design

Some method description should be expanded and explained. Also, the measurements of epicardial activation patterns does not allow for a true measurement of conduction velocity. The data however with some modification of text equally supports the conclusions made, ie that normal conduction patterns are restored by EB1 overexpression.

Validity of the findings

The data presented support a role for EB1 in the arrhythmias of the reported mode. One reason may be the preservation of Cx43, however as indicated by my specific comments other important effects may occur downstream of EB1. The study does not show causality in relation to Cx43 and discussion and conclusions should reflect this. In places, claims are made that needs substantiation (see specific comments).

Additional comments

L60: The given reference does not mention hypothermia and it remains unclear what role if any hypothermia plays and what relation the experiments presented has to CPB (see below).
L63: these refs are a review and a specific reference of recent date, which are hardly the first to demonstrate the role of conduction disturbances. I would suggest some background research and a reference to either original breakthroughs or a major textbook or review on the subject.
L66: The chamber distribution of cardiac connexins was established a long time before the quoted papers. I would suggest the authors to revisit the literature inspiration may be found in pmids 15094343 and 8534903
L71 the remodeling concept is much older than the references given. Eg work by Peters and Severs, etc.
L75 I would rather expect increased lateral coupling to reduce anisotropy, but it has mainly been suggested that lateralized connexins are not fuctional although this remains to be established.
L91 There is also evidence that Cx43 may anchor EB1 and modulate the transport of NaV1.5 to the membrane. Any loss of NaV1.5 forward trafficking will likely also affect conduction, which is has major implication for the interpretation of the presented results. This should be included in the introduction and discussion of the subject. See pmids 25139742 and 34092082.
L135 With which gas was the solution equilibrated? Here it sounds like pure oxygen, but that would shift the pH of the solution?
L137 what is the nature of the ischemic insult, global no flow ischemia, perfusion of solution equilibrated with N2, or?
L148 do you mean that successful experiments were performed on the hearts of all the animals subjected to the procedure, or what does modeled mean in this context?
L154 MEA technique. Apparently, no pacing is performed from the MEA, which means that you measure break-through patterns and not conduction as such. This makes it impossible to calculate the conduction velocity (CV) and you should probably restrict the analysis to total activation times and activation time heterogeneity. Also, some more explanation should be provided about sampling, filtering and analysis. In particular, a description of the inhomogeneity calculations is needed.
L173 please provide product numbers for antibodies here and elsewhere.
L204 what is the content of the lysis buffer
L271 are the data for EB1 and Cx43 normalized for tubulin and Na/K-ATPase or not? The ATPase should be specified. Did the EB1 and Cx43 levels correlate in each animal?
L272 what did the experiments show in relation to conduction and arrhythmia? Did this correlate with the EB1 and Cx43 levels?
L275 what is the point of the red arrow in figure?
L228 fig 2A?
L295 I would suggest reduced instead of improved in the wording, since improved requires some interpretation.
L301 Does RA refer to a lead? The sentence seems unfinished.
L307 Maybe a bit premature to conclude about causality in the title
L318 electrical conduction activation, should probably be ‘electrical activation’.
L321 ‘Considering…’ This sentence seems superfluous and reference unnecessary, since the role of conduction in arrhythmias were hardly discovered in 2024.
L325 As noted under methods, you are not measuring CV but rather some reflection of the AP conduction/distribution by the underlying myocardial structures, including the Purkinje network. This also explains the very high ‘velocities’ that you report. Your point may, however, be equally made just using the total activation times for the field. It would also seem that the data in B-D are the same as in E, one representation will suffice (all comparisons can be made in E).
L328 It is unclear from the methods how inhomogeneity was calculated. The maps seem highly correlated to the activation maps, so what exactly does this add? Also, B-D seems repeated in H and E-G in I. One representation should suffice as for figure 3.
L331 T2 is actually not significantly different from T0 in the I/R group.
L333 Do they add any independent information in reality? If not the homogeneity numbers could be added to figure 3.
L334 Is ‘index’ missing in this sentence? Samme for L336
L348 why not use a 2way anova since you make comparisons within and between datasets. Did you make any corrections for multiple comparisons when using the t-test and what comparisons were made? These considerations should be applied here and elsewhere relevant.
L365 I reckon you wanted to test if EB1 overexpression increases/rescues Cx43, in its present form the sentence concludes that this will be so.
L368 what measures were taken to ensure equal illumination and sampling of the images? This is important for any quantification of intensities from the images
L369 with vertically, you mean longitudinally (along the long axis of the myocytes)? What do you mean ‘tended’ to localize with N-cad? Some zooms could be shown and indices of colocalization.
L372 and 372 figure 5A and B? C is polymeric tubulin.
L375 5B?
L378 as well as a multitude of other proteins including NaV1.5
L394 these experiments do not test whether a rescue of Cx43 at the ID prevents arrhythmia, rather they test if EB1 overexpression rescues Cx43 at the ID.
L396 some description of the changes in IR and con group could be made and then contrasted to the EB1 overexpression. It is not clear what ‘abundant distribution' means.
L403 I reckon that the N-cad shows the ID location, whether the Cx43 staining is GJ (although most likely) it may as well be undocked hemichannels.
L414 immunofluorescence misspelled
L416 probably better to use membrane Cx43 throughout
L422 The data correlates Cx43 to arrhythmia, but that does not prove causality. If microtubules are destabilized, a host of other membrane proteins including ion channels and transporters could be affected. Evidence along these lines has been shown for NaV1.5 and a thorough search for other candidates should be conducted. Any of these changes or their combination, may be underlying the arrhythmias seen.
L436 Cx43 may not be the only target, NaV1.5 has been shown to be affected and other ID proteins transported along microtubules should be considered. A discussion of this should be included and any conclusions regarding the causality of Cx43 in causing arrhythmias should be modified accordingly. This applies to many places in the text and I will not comment on this further.
L451 again I would encourage reduced rather than improved. Also, since you do not measure conduction velocity, I would probably use activation patterns and heterogeneity instead.
L457 Conversely, Cx43 may also be essential for guiding in EB1 as first shown by Francis et al 22022608 and later shown for cardiomyocytes by the Delmar group (pmid given earlier). This may induce some element of reverse causation.
L461 EB1 concentration at the ID was not shown (statement of ‘tended’) and should be substantiated.
L473 conduction disturbances are likely involved in reentrant VT/VF making a good case for Cx43, In contrast, other factors are more likely underlying PVBs and it is unlikely that reduced Cx43 should lead to AVB. Rather the reduction in all arrhyutmic events points to more complex effects of EB1 overexpression. In light of this, the following claim that ‘GJs are the key factor to be considered’ makes no sense.
L500 the study does not show causality and conclusions should be modified to reflect this.
L501 not sure I understand why it ‘further promotes…’ isn’t this the only proposed mechanism?

---

## Round 0.2 · Major Revisions

Please address remaining concerns of the reviewers and amend manuscript accordingly

Reviewer 1 ·

Basic reporting

The author has addressed the referee's comments and made the required changes; however, there are still minor composition and grammatical errors that need to be refined.

1. Abstract
-The methods are still not properly written.

Suggestion: 4-6 week old male Sprague-Dawley (SD) rats were randomly assigned to 4 groups with a control group receiving no treatment. In the treatment groups, the rats received an injection of a negative control adenovirus (AAV9-CON) or an adenoviral vector containing Mapre1 gene (AAV9-EB1) or an equal volume of saline via the tail vein. After four weeks, untreated rat hearts underwent continuous isolated heart perfusion for 5 minutes, while the treatment groups were subjected to Langendorff isolated heart ischemia-reperfusion.

2. Material

Line 202: 2.2 Ex vivo electrophysiological recordings
Line 190: change the subheading to ‘Animal experimental design’
Line 227-242 : combine 2.3-2.4

3. Resuts

Figure 3 (B-D) label the graphs with T0, T1, T2
Figure 4 (B-G) label the graphs with T0, T1, T2

Experimental design

No comment

Validity of the findings

No comment

Reviewer 2 ·

Basic reporting

The authors have addressed all comments accordingly.
There are a few small grammar/formatting comments that still require revision.

L151-152: Please correct grammar for Sentence "If......". Perhaps consider using past conditionals?

L163: Please remove “immunohistochemical” as this term is incorrect. Provide the long term for HE, which is 'Hematoxylin-Eosin'.

L216: Please rectify incorrect spelling. "Real-Time".

L245: Too many “or”, suggest separate with commas then add “or” before the last word.

L354-355: Please correct grammer for sentence "Next, we...." to align with the past tense used in following sentences.

L385-387: Consider removing the terms "exhibited concentrated distribution of...". To ensure clarity, the sentence could be modified as such-- to include the following "Cx43 is more concentrated at the GJs" (give and take).

Experimental design

Revisions were made accordingly.

Validity of the findings

Revisions were made accordingly. Below are replies to the authors' rebuttal:

1) "We recorded the HR at the end of 30 min of stabilization and 15 and 30 min after reperfusion (T0, T1, and T2). Arrhythmias do not necessarily occur precisely at the time points T0, T1, and T2. The arrhythmias we report may not occur at these specific times. In fact, arrhythmias primarily happen within the first 15 minutes after ischemia-reperfusion. By the time of T1, arrhythmias are already rare."
Reply:
This is an important finding. To avoid reader's misunderstanding, pplease include this explanation in section 3.2. (L275-287) or 3.3. (L298-340). Alternatively, the authors could include this in the discussion.

2) "We referenced some other papers(PMID: 22406144), which did not include separate images of the nuclear fluorescence. Considering the reduction of printing costs and aiming for a more intuitive and simpler presentation, we have chosen not to include separate images of the nuclear fluorescence. However, we will indicate this in the figure legend. Do you think this is acceptable? If you do not recommend this approach, we can add them back in."
Reply:
Current images are great. However I highly suggest to include the marker names in the IF images. There is a way without compromising space, please see example in Figure 3 of the following publication: https://pubmed.ncbi.nlm.nih.gov/29805478/.

Additional comments

I highly appreciate the authors' effort in addressing every comment.
I highly recommend this article to be accepted after minor revisions are addressed.

Reviewer 3 ·

Basic reporting

I have now reviewed the comments and changes made by the authors. Unfortunately, some essential comments have not been adequately addressed.

In particular, the authors still claim to measure conduction velocity, which is simply not possible unless you pace inside the MEA or immediately next to it. Since no mention is made of pacing, I conclude that the measurements are made on normal intrinsic beats and thus the maps show break-through of waves coming from the underlying ventricular tissue. Inspection of the maps in figure 3 support this and shows vast expanses with identical activation times. Consequently the velocity in these areas is near infinite, which obviously makes no sense. Pasting the manual into the response does not change this problem and besides it has no mention of conduction velocity. Please remove the wording ‘conduction velocity’ when referring to these data. The heterogeneity of activation may have some information, but please explain how this was determined (not just referring to the software).

Given the evidence that the microtubular network is essential for trafficking of a large number of membrane proteins, it should be discussed what other mechanisms could be underlying the results. The discussion and conclusion should also acknowledge the limitation of just focusing on one effector, Cx43, when other players may be involved. Along these lines Cx43 is unlikely to be involved in AVB and its prevention, I did not see any changes that brought this to light. For this whole issue, just acknowledging that NaV1.5 also depends on EB1 in the introduction is really not enough.

Some further points in order of the comments:
I still find the choice of references peculiar. Normal practice would be to refer those who originally provided evidence for the claims made. I fail to see how ‘recent’ qualifies in itself. Some (maybe all) new references refers to author first names.
With respect to enhanced transverse conduction, it will reduce anisotropy. Not induce or increase it.
Did you turn of the perfusion during ischemia, equilibrate the KH solution with N2 or? Please describe the procedure.
With respect to statistics, what corrections if any was made for multiple comparisons? This should be included in the description of statistics.

Experimental design

please refer to the above

Validity of the findings

please refer to the above

---

## Round 0.3 · Major Revisions

As you can see, two reviewers are satisfied by the revision, but the reviewer #3 finds serious issues with research design and indicates that your claim of measuring conduction velocity is incorrect. Although this reviewer recommended rejection, I decided to give you another chance to address their concern and amend manuscript accordingly.

Reviewer 1 ·

Basic reporting

no comment

Experimental design

no comment

Validity of the findings

no comment

Additional comments

The modifications have improved the structure, presentation, and analysis within the manuscript, making the study more comprehensive and easier to understand. The author has provided additional explanations where necessary and refined the methodology to ensure clarity and precision in the research process. The revised manuscript now presents a clear and valuable contribution to the field, adding to the understanding of the subject matter. With the improvements made based on the reviewer's suggestions, the study is now considered to be of sufficient quality to be published.

Reviewer 2 ·

Basic reporting

The authors have addressed all comments accordingly.

Experimental design

Revisions were made accordingly.

Validity of the findings

Revisions were made accordingly.

Additional comments

The authors have incorporated all feedback and implemented the suggested revisions. I recommend accepting this manuscript.

Reviewer 3 ·

Basic reporting

The authors have addressed most of my concerns; however, the issue on conduction velocity persists. It is simply not possible to measure a real conduction velocity if you are not sure that you measure perpendicular to the direction of the wavefront. This condition is rarely if ever met in epicardial measurements of spontaneous beats. The result is overestimates as evidenced by your extremely high ‘velocities’. The heterogeneity estimates and total activation times of your maps could equally support the concept that conduction has changed.

Regarding: Quantification of Activation Heterogeneity
This clarifies the measurements of inhomogeneity, but it will not be clear to the reader in the present form of the manuscript. A reference to the paper of Wim et al would fix this.

Regarding: Conduction Velocity
You can fit a parabolic surface to break through data and estimate an apparent velocity, but only in few cases will it correspond to the real conduction velocity, which can only be measured when you stimulate in the plane of measurements (eg the Wim et al paper). The paper you refer to states this quite clearly: ‘The difficulty is that the direction of propagation must be accurately known. If the wavefront is not perpendicular to the line connecting two electrodes, but instead is approximately parallel, the two sites will activate nearly simultaneously. Since velocity is computed from delta x/delta t, artificially small values of delta t lead to severe overestimates of velocity (Fig. 1).’ Even with the simulation, they find values that are more than double those measured from an intra-electrode array pacing, which completely recapitulates the problem. They go on to conclude that: ‘When applied to epicardial data from ventricular rhythms, the velocity vector field corresponds to the velocity of the projection of the three-dimensional (3-D) wavefront onto the 2-D mapping surface. Hence, one should be careful when interpreting speed estimates if a strong transmural component is likely to be present.’ Since the wave front propagates from endo- to epicardium, there is a strong transmural component and as a consequence, you are not measuring conduction velocity.

Regarding: anisotropy
I never disputed that ischemia can increase anisotropy, however, the redistribution of connexins to the lateral sides does not explain this if they are functional. In isolation, this would lower longitudinal conduction more than transverse and thus reduce anisotropy. It’s a minor point that does not really affect the main message of the manuscript.

Regarding: ischemia
But did you stop the perfusion while ‘storing’ it? If so, it should be added to the methods.

Experimental design

Please see text above

Validity of the findings

Please see text above

Additional comments

Please see text above

---

## Round 0.4 · accepted · Accept

All issues pointed out by the reviewers were adequately addressed and the revised manuscript is acceptable now.

Reviewer 3 ·

Basic reporting

The authors have now addressed my concerns and I have no further comments.

Experimental design

see above

Validity of the findings

see above

Additional comments

see above